# Anti-β2-glycoprotein I/HLA-DR Antibody and Adverse Obstetric Outcomes

**DOI:** 10.3390/ijms241310958

**Published:** 2023-06-30

**Authors:** Kenji Tanimura, Shigeru Saito, Sayaka Tsuda, Yosuke Ono, Hajime Ota, Shinichiro Wada, Masashi Deguchi, Mikiya Nakatsuka, Takeshi Nagamatsu, Tomoyuki Fujii, Gen Kobashi, Hisashi Arase, Hideto Yamada

**Affiliations:** 1Department of Obstetrics and Gynecology, Kobe University Graduate School of Medicine, Kobe 650-0017, Japan; taniken@med.kobe-u.ac.jp (K.T.); deguchi@med.kobe-u.ac.jp (M.D.); 2Department of Obstetrics and Gynecology, University of Toyama, Toyama 930-0194, Japan; s30saito@med.u-toyama.ac.jp (S.S.); syk3326@med.u-toyama.ac.jp (S.T.); 3Department of Obstetrics and Gynecology, University of Yamanashi, Yamanashi 409-3898, Japan; nadal.babolat@hotmail.co.jp; 4Department of Obstetrics and Gynecology, Teine Keijinkai Hospital, Sapporo 006-8555, Japan; ota-ha@keijinkai.or.jp (H.O.); 6200@keijinkai.or.jp (S.W.); 5Faculty of Health Sciences, Okayama University, Okayama 700-8558, Japan; mikiya@cc.okayama-u.ac.jp; 6Department of Obstetrics and Gynecology, The University of Tokyo, Tokyo 113-8655, Japan; tnag-tky@umin.ac.jp (T.N.); fujiit-tky@umin.org (T.F.); 7Department of Obstetrics and Gynecology, International University of Health and Welfare Narita Hospital, Narita 286-8520, Japan; 8Sanno Hospital, Tokyo 107-0052, Japan; 9Department of Public Health, Dokkyo Medical University, Tochigi 321-0293, Japan; genkoba@dokkyomed.ac.jp; 10Department of Immunochemistry, Research Institute for Microbial Disease, Osaka University, Suita 565-0871, Japan; arase@biken.osaka-u.ac.jp; 11Center for Recurrent Pregnancy Loss, Teine Keijinkai Hospital, Sapporo 006-8555, Japan

**Keywords:** antiphospholipid syndrome, autoantibody, β2-glycoprotein I, fetal growth restriction, HLA class II, hypertensive disorders of pregnancy, preterm delivery, recurrent pregnancy loss

## Abstract

Anti-β2-glycoprotein I/HLA-DR (anti-β2GPI/HLA-DR) antibody has been reported to be associated with antiphospholipid syndrome and recurrent pregnancy loss (RPL). We conducted a prospective multicenter cross-sectional study aimed at evaluating whether the anti-β2GPI/HLA-DR antibody is associated with adverse obstetric outcomes and RPL. From 2019 to 2021, serum anti-β2GPI/HLA-DR antibody levels (normal, <73.3 U) were measured in 462 women with RPL, 124 with fetal growth restriction (FGR), 138 with hypertensive disorders of pregnancy (HDP), 71 with preterm delivery before 34 gestational weeks (preterm delivery (PD) ≤ 34 GWs), and 488 control women who experienced normal delivery, by flow cytometry analysis. The adjusted odds ratios (aORs) of anti-β2GPI/HLA-DR antibody positivity for adverse obstetric outcomes and RPL were evaluated on the basis of comparisons between the control and each patient group, using multivariable logistic regression analysis. The following were the positivity rates for the anti-β2GPI/HLA-DR antibody in the patient and control groups: RPL, 16.9%; FGR, 15.3%; HDP, 17.4%; PD ≤ 34 GWs, 11.3%; and the control, 5.5%. It was demonstrated that anti-β2GPI/HLA-DR antibody positivity was a significant risk factor for RPL (aOR, 3.3 [95% confidence interval {CI} 1.9–5.6], *p* < 0.001), FGR (2.7 [1.3–5.3], *p* < 0.01), and HDP (2.7 [1.4–5.3], *p* < 0.01) although not for PD ≤ 34 GWs. For the first time, our study demonstrated that the anti-β2GPI/HLA-DR antibody is involved in the pathophysiology underlying FGR and HDP, as well as RPL.

## 1. Introduction

Previous studies have demonstrated that misfolded proteins generated in the endoplasmic reticulum (ER), which are generally eliminated by ER-associated degradation (ERAD), can be rescued from ERAD and transported to the cell’s surface without being processed into peptides. This process can occur in the ER through an association between the misfolded proteins and the peptide-binding groove of human leukocyte antigen (HLA) class II molecules [1]. Furthermore, misfolded proteins complexed with HLA class II molecules of disease-susceptible alleles serve as autoantibody targets, and these complexes are involved in the pathogenesis of several autoimmune diseases. For example, immunoglobulin (Ig) G heavy chain/HLA-DR complexes in patients with rheumatoid arthritis (RA), myeloperoxidase/HLA-DR complexes in patients with microscopic polyangiitis, and thyroid-stimulating hormone receptor/HLA-DP complexes in patients with Graves’ disease were major targets for autoantibodies [2,3,4]. Recently, it was also reported that not only misfolded proteins but also DNA can form complexes with HLA class II molecules, and that DNA/HLA-DR complex is involved in systemic lupus erythematosus (SLE) pathogenesis [5]. 

Antiphospholipid syndrome (APS) is an autoimmune disease characterized by the presence of clinical manifestations including vascular thrombosis and pregnancy morbidity and by the presence of antiphospholipid antibodies (aPLs) [6]. The following are pregnancy morbidities included in the diagnostic criteria for APS [6]: unexplained fetal death beyond 10 gestational weeks (GWs); premature births of normal neonates before 34 GWs due to eclampsia, severe eclampsia, or recognized features of placental insufficiency; and recurrent pregnancy loss (RPL). aPLs mainly recognize β2-glycoprotein I (β2GPI), a phospholipid-binding protein [7,8]. Generally, aPLs are detected by enzyme-linked immunosorbent assay (ELISA) methods, using plates containing solid-phase negatively charged phospholipids or plates with β2GPI [9,10]. Recently, we have observed that the autoantibody against β2GPI/HLA-DR complexes (anti-β2GPI/HLA-DR antibody) is involved in APS pathogenesis [11]. Furthermore, as the antibody can recognize unique epitopes that are not recognized by conventional aPLs, our previous studies have demonstrated that anti-β2GPI/HLA-DR antibody measurements may improve the sensitivity of diagnostic testing for APS [11,12]. Additionally, we have noted that among 227 women with RPL, 22.9% of them were positive for anti-β2GPI/HLA-DR antibody and, interestingly, 19.8% of 121 women with unexplained RPL were positive for the antibody [12]. However, no studies have addressed whether the anti-β2GPI/HLA-DR antibody can be a significant risk for other adverse obstetric outcomes, including fetal growth restriction (FGR), hypertensive disorders of pregnancy (HDP), and preterm delivery (PD).

In this prospective multicenter cross-sectional study, we investigated the association between the anti-β2GPI/HLA-DR antibody and adverse obstetric outcomes, as well as RPL.

## 2. Results

### 2.1. Anti-β2GPI/HLA-DR Antibody in Women with RPL

The clinical characteristics and anti-β2GPI/HLA-DR antibody prevalence in 462 women with RPL and 488 control women enrolled in this multicenter study are presented in Table 1. Autoimmune diseases of the 34 women in the RPL group were as follows: APS, *n* = 13; Graves’ disease, *n* = 7; SLE, *n* = 5; Sjögren syndrome (SJS), *n* = 4; RA, *n* = 3; dermatomyositis, *n*=1; and autoimmune hepatitis, *n* = 1. The RPL group had significantly higher maternal age, gravidity, number of prior spontaneous miscarriages and prior stillbirths at ≥22 GWs, and proportion of women with autoimmune diseases and women who used immunosuppressive drugs than did the control group. Conversely, maternal body mass index (BMI) and parity in the RPL group were significantly lower than those values in the control group. The proportions of women who had a history of thrombosis, chronic hypertension, or a smoking habit were not different between the two groups. Notably, the proportion of women who were positive for anti-β2GPI/HLA-DR antibody in the RPL group (16.9%, *p* = 4.4 × 10^−8^) was significantly higher than that of the control group (5.5%). 

Risk factors for RPL among 462 women with RPL and the association between anti-β2GPI/HLA-DR antibody and these factors are shown in Figure 1. The following were the incidences of risk factors for 462 women with RPL: uterine malformation, 5.4%; thyroid dysfunction, 14.8%; chromosomal abnormality in women with RPL and their spouses, 3.0%; aPL positivity, 14.7%; low factor XII activity, 8.5%; low protein C activity, 2.2%; low protein S activity, 13.4%; and unexplained, 53.0%.

The comparison of clinical characteristics between 78 women with RPL who were positive for anti-β2GPI/HLA-DR antibody and the remaining 384 patients who were negative for the autoantibody is shown in Table 2. The proportions of patients in the anti-β2GPI/HLA-DR antibody-positive group who had autoimmune diseases, who used immunosuppressive drugs, and who were positive for aPLs were significantly higher than those in the anti-β2GPI/HLA-DR antibody-negative group. The other factors were not different between the two groups.

### 2.2. Anti-β2GPI/HLA-DR Antibody in Women with FGR

The clinical characteristics and anti-β2GPI/HLA-DR antibody prevalence in 124 women with FGR, including 49 and 83 during their previous and current pregnancies, respectively, as well as 488 control women enrolled in this study are presented in Table 3. Autoimmune diseases of the seven women in the FGR group were as follows: APS, *n* = 3; SLE, *n* = 2; and SJS, *n* = 2. Maternal age was significantly lower and the proportion of women who used immunosuppressive drugs was significantly higher in women who had FGR during their current pregnancies than in the control group. Women who had FGR during their previous pregnancies had significantly higher numbers of gravidity, parity, and prior stillbirths at ≥22 GWs than did those in the control group. Compared with the control group, the number of prior spontaneous miscarriages, the proportion of patients with autoimmune diseases and the number with chronic hypertension were significantly higher in both women who had FGR during their previous pregnancy and those who had FGR during their current pregnancies. Maternal BMI and the proportions of women who had a history of thrombosis, a smoking habit or pregnancies following assisted reproductive technology (ART) were not different between either the group of patients who had FGR during their previous pregnancy or the group of patients who had FGR during their current pregnancy and the control group. The proportions of women who were positive for anti-β2GPI/HLA-DR antibody in the group of women who had FGR during their previous pregnancies (18.4%, *p* = 0.003), the group who had FGR during their current pregnancies (14.5%, *p* = 0.006), and the group who had FGR during their previous or current pregnancies (all FGR groups) (15.3%, *p* = 0.0005) were significantly higher than that of the control group (5.5%). 

### 2.3. Anti-β2GPI/HLA-DR Antibody in Women with HDP

The clinical characteristics and anti-β2GPI/HLA-DR antibody prevalence in 138 women with HDP, including 64 and 79 during their previous and current pregnancies, respectively, as well as 488 control women, are shown in Table 4. Autoimmune diseases of the nine women in the HDP group were as follows: APS, *n* = 3; SLE, *n* = 3; Graves’ disease, *n* = 2; and pemphigus foliaceus, *n* = 1. Maternal age, the number of prior spontaneous miscarriages, and the proportions of patients with autoimmune diseases and those with chronic hypertension in both women who had HDP during their previous pregnancies and those during their current pregnancies were significantly higher than those in the control group. Women who had HDP during their previous pregnancies had significantly higher gravidity than did those in the control group. The group of women who had HDP during their current pregnancies had significantly higher maternal BMI and proportions of patients who used immunosuppressive drugs and who had a smoking habit than did the control group. Parity, the number of prior stillbirths at ≥22 GWs, and the proportion of women who had a history of thrombosis and pregnancies following ART were not different between the group of patients who had HDP during their previous pregnancy or the group of patients who had HDP during their current pregnancies and the control group. The proportions of women who were positive for anti-β2GPI/HLA-DR antibody in the group of women who had HDP during their previous pregnancies (18.8%, *p* = 0.0007), the group who had HDP during their current pregnancies (16.5%, *p* = 0.001), and the group who had HDP during their previous or current pregnancies (all HDP groups) (17.4%, *p* = 2.0 × 10^−5^) were significantly higher than that of the control group (5.5%).

### 2.4. Anti-β2GPI/HLA-DR Antibody in Women with PD before 34 GWs

The clinical characteristics and anti-β2GPI/HLA-DR antibody prevalence in 71 women with PD before 34 GWs (PD ≤ 34 GWs), including 35 and 39 during their previous and current pregnancies, respectively, as well as 488 control women, are shown in Table 5. Autoimmune diseases of the five women in the PD ≤ 34 GWs group were as follows: APS, *n* = 1; SLE, *n* = 2; SJS, *n* = 1; and RA, *n* = 1. Gravidity and the proportion of patients with autoimmune diseases in both women who had PD ≤ 34 GWs during their previous and those who had PD ≤ 34 GWs during their current pregnancies were significantly higher than those in the control group. Women who had PD ≤ 34 GWs during their previous pregnancies had significantly higher maternal age, parity, and proportions who used immunosuppressive drugs and who had a chronic hypertension than did those in the control group. The group of women who had PD ≤ 34 GWs during their current pregnancies had a significantly higher number of previous spontaneous miscarriages than did those in the control group. Maternal BMI, number of prior stillbirths at ≥22 GWs, and the proportions of women who had a history of thrombosis, a smoking habit, or pregnancies following ART were not different between the group of patients who had PD ≤ 34 GWs during their previous pregnancy or the group of patients who had PD ≤ 34 GWs during their current pregnancies and the control group. The proportions of women who were positive for anti-β2GPI/HLA-DR antibody in the group of women who had PD ≤ 34 GWs during their previous pregnancies (14.3%, *p* = 0.05), the group who had PD ≤ 34 GWs during their current pregnancies (7.7%, *p* = 0.4), and the group who had PD ≤ 34 GWs during their previous or current pregnancies (all PD ≤ 34 GWs groups) (11.3%, *p* = 0.06) were not different from that of the control group (5.5%).

### 2.5. Associations between Anti-β2GPI/HLA-DR Antibody and Adverse Obstetric Outcomes

Crude ORs (cORs) and adjusted ORs (aORs) of anti-β2GPI/HLA-DR antibody positivity for RPL, FGR, HDP, and PD ≤ 34 GWs are shown in Table 6. Univariate logistic regression analysis showed that anti-β2GPI/HLA-DR antibody positivity was associated with increased odds of RPL (cOR, 3.5; 95% confidence interval [CI], 2.2–5.5), FGR in previous pregnancies (cOR, 3.8; 95% CI, 1.7–8.7), FGR in current pregnancies (cOR, 2.9; 95% CI, 1.4–6.0), HDP in previous pregnancies (cOR, 3.9; 95% CI, 1.9–8.2), HDP in current pregnancies (cOR, 3.4; 95% CI, 1.7–6.8), and PD ≤ 34 GWs in previous pregnancies (cOR, 2.8; 95% CI, 1.02–7.9). Furthermore, multivariable logistic regression analysis showed that anti-β2GPI/HLA-DR antibody positivity was associated with increased odds of RPL (aOR, 3.3; 95% CI, 1.9–5.6), FGR in previous pregnancies (aOR, 3.1; 95% CI, 1.2–8.0), FGR in current pregnancies (aOR, 2.6; 95% CI, 1.2–5.6), HDP in previous pregnancies (aOR, 2.5; 95% CI, 1.01–6.2), and HDP in current pregnancies (aOR, 2.7; 95% CI, 1.2–6.0). However, anti-β2GPI/HLA-DR antibody positivity was not associated with PD ≤ 34 GWs, either in previous pregnancies or in current pregnancies.

## 3. Discussion

This prospective multicenter cross-sectional study has shown for the first time that the anti-β2GPI/HLA-DR antibody is a significant risk factor for FGR (aOR, 2.7; 95% CI, 1.3–5.3; *p* < 0.01) and HDP (aOR, 2.7; 95% CI, 1.4–5.3; *p* < 0.01), as well as RPL (aOR, 3.3; 95% CI, 1.9–5.6; *p* < 0.001). 

In our previous study, immunohistochemistry and in situ proximity ligation assay, which detects proximity (<40 nm) between two molecules [13], revealed that β2GPI and HLA-DR had colocalized in endothelial cells of the vessels in the placental decidua obtained from patients with APS with spontaneous miscarriages [11]. Furthermore, we noted that a human aPL monoclonal antibody obtained from a patient with APS (EY2C9) [14] exhibited complement-mediated cytotoxicity against β2GPI/HLA-DR complex-expressing cells, although neither β2GPI-expressing cells nor HLA-DR expressing cells [11]. Based on these results, it is speculated that the anti-β2GPI/HLA-DR antibody may damage β2GPI/HLA-DR complex-expressing vascular endothelial cells in the placental decidua, and that the vascular endothelial damage can lead to placental insufficiency, and may eventually cause FGR, HDP, and RPL.

Conversely, in our previous study, we reported that 22.9% of 227 women with RPL were positive for the anti-β2GPI/HLA-DR antibody and that 19.8% of 121 women with unexplained RPL possessed this autoantibody [12]. In that study, anti-β2GPI/HLA-DR antibody positivity was the most frequently observed risk factor for RPL, except for unexplained RPL. In the present study, 78 (16.9%) of 462 women with RPL were positive for anti-β2GPI/HLA-DR antibody, and 39 (15.9%) of 245 women with unexplained RPL tested positive for the autoantibody. In the present study, anti-β2GPI/HLA-DR antibody positivity was the most frequently observed risk factor for RPL, and the results are similar to those of our previous study. Furthermore, the present study has shown for the first time that anti-β2GPI/HLA-DR antibody positivity is a significant risk factor for RPL, using multivariable logistic regression analyses adjusting for age, maternal BMI, and smoking habit. Additionally, the proportions of patients with RPL who had autoimmune diseases and aPLs in the anti-β2GPI/HLA-DR antibody-positive population were significantly higher than those in anti-β2GPI/HLA-DR antibody-negative population. Patients who have anti-β2GPI/HLA-DR antibodies may possess immune disorders.

On the other hand, some patients in the groups of RPL, FGR, HDP, and PD ≤ 34 GWs used immunosuppressive drugs for treatments of underlying autoimmune diseases. All of them were treated with oral prednisolone (PSL), and a few of them were treated with oral PSL combined with another immunosuppressant (i.e., azathioprine, tacrolimus, etc.). However, it is unclear whether these immunosuppressive agents influence the production of anti-β2GPI/HLA-DR antibodies. The number of patients who used immunosuppressive drugs in this study was small; therefore, we believe that the results of this study were not influenced by the use of these drugs.

The FGR and HDP groups had significantly higher anti-β2GPI/HLA-DR antibody positivity rates than did the control group; however, that in the group of patients who had PD ≤ 34 GWs was not different from the control group. Similarly, univariate and multivariable logistic regression analyses revealed that anti-β2GPI/HLA-DR antibody positivity was a significant risk factor for FGR and HDP, although not for PD ≤ 34 GWs. In the present study, 57 (80.3%) of all 71 patients with PD ≤ 34 GWs had spontaneous or artificial preterm deliveries following tocolysis failure. It is believed that placental dysfunction and maternal endothelial dysfunction cause pre-eclampsia and FGR [15,16]. Conversely, it is believed that up to 40–50% of PDs are associated with microbial etiologies [17,18]. Therefore, if the only patients enrolled in this study who had PD ≤ 34 GWs were associated with HDP and/or FGR caused by placental dysfunction, it is possible that anti-β2GPI/HLA-DR antibody positivity was also associated with PD ≤ 34 GWs. 

PD ≤ 34 GWs due to the recognized features of placental insufficiency, including HDP, and FGR, is one of the clinical symptoms of APS. Therefore, APS can be closely associated with HDP and FGR pathogeneses. In our previous study, we observed that 83% of 120 patients with APS had anti-β2GPI/HLA-DR antibodies; however, approximately 50% of patients with APS who had anti-β2GPI/HLA-DR antibodies were negative for IgG anti-cardiolipin (aCL) or IgG anti-β2-glycoprotein I antibody (aβ2GPI) [11]. In another study, we noted that 67.3% of 52 women with RPL who were positive for anti-β2GPI/HLA-DR antibody did not have any aPLs, which are included in the diagnostic criteria for APS [12]. Additionally, 27% of 111 patients with idiopathic chronic limb ulcers who were negative for conventional aPLs had anti-β2GPI/HLA-DR antibodies [19]. β2GPI is the main phospholipid-binding molecule recognized by aPLs [7,8]. Circular β2GPI is converted to linear β2GPI, which exposes the major epitope for aPLs when β2GPI binds to ionic phospholipids [20]. Therefore, aPLs are detected using ELISA in clinical settings using plates containing solid-phase negatively charged phospholipids, that is, cardiolipin and β2GPI, or negatively charged plates containing β2GPI [9,10]. Conversely, it has been observed that β2GPI/HLA-DR complexes express epitopes shared by β2GPI/cardiolipin complexes or β2GPI binding to negatively charged plates as well as unique epitopes that are not present on such complexes [11]. Therefore, it is possible that there are patients with HDP, FGR, and RPL caused by similar pathologies to APS, but who cannot be diagnosed by aPL detection that meets the diagnostic criteria for APS. Indeed, 71 (91.0%) of 78 anti-β2GPI/HLA-DR antibody-positive women with PRL, 17 (89.5%) of 19 antibody-positive women with FGR, and 22 (91.7%) of 24 antibody-positive women with HDP were not diagnosed with APS. This novel autoantibody against β2GPI/HLA-DR complexes may provide new insights into pathogeneses of HDP, FGR, and RPL which have not been recognized thus far.

The combination therapy of heparin and low-dose aspirin (LDA) is the standard treatment for pregnant women with APS [21]. Among women with RPL who were positive for anti-β2GPI/HLA-DR antibody but negative for conventional aPLs, the live-birth rate was higher in the group of patients who received LDA or LDA-plus-heparin therapy than in the group of patients treated with neither LDA nor heparin (unpublished observation). LDA or LDA-plus-heparin therapy may be effective for anti-β2GPI/HLA-DR antibody-positive patients with FGR, HDP, or RPL.

The present study has some limitations. First, this study is not a prospective cohort study but a prospective cross-sectional study; therefore, we could not prove the accurate pathogenicity of the anti-β2GPI/HLA-DR antibody. Presently, a cohort study, in which normal pregnant women undergo serum anti-β2GPI/HLA-DR antibody level measurements in the first trimester and they are followed up with for the occurrence of adverse obstetric outcomes, is already being undertaken. Second, in the present study, HDP was defined as hypertension during pregnancy. Strictly speaking, the clinical criteria of APS include PD ≤ 34 GWs due to eclampsia or preeclampsia (PE), but our present study included only 35 women with early-onset (EO)-PE during their previous and/or current pregnancies. Six of them were positive for the anti-β2GPI/HLA-DR antibody, and the positivity rate of the autoantibody in women with EO-PE was significantly higher than that in the control group (17.1% vs. 5.5%, *p* = 0.02), but the number of patients with EO-PE was too small. Therefore, further studies enrolling patients with EO-PE only are needed. In addition, HDP is classified into four categories (i.e., chronic hypertension, gestational hypertension, PE, and chronic hypertension with superimposed PE) [22,23], but in this study, we could not assess whether the anti-β2GPI/HLA-DR antibody is associated with the differences in the disease type and severity of HDP. Third, in the diagnostic criteria for APS, positive results from aPLs testing using two tests 12 months or more apart are required. However, repeat testing to confirm persistent positive results for the anti-β2GPI/HLA-DR antibody was not performed in this study. Fourth, the mechanisms by which the anti-β2GPI/HLA-DR antibody leads to HDP, FGR, and RPL are unclear; therefore, to resolve the enigmas, not only clinical studies but also basic studies are needed. Moreover, further studies are required to confirm the conclusions of this study.

However, recently, it has been reported that the anti-β2GPI/HLA-DR antibody is a significant risk factor for endometriosis in infertile women and for recurrent implantation failure in infertile women treated with ART [24]. The results of these studies on anti-β2GPI/HLA-DR antibodies will help us better understand the pathogeneses of not only adverse obstetric outcomes, including HDP, FGR, and RPL, but also infertility, and develop treatments for such disorders.

## 4. Materials and Methods

This prospective multicenter cross-sectional study was approved by the review boards of five medical centers (reference number 190102 at Kobe University Hospital), and written informed consent was obtained from all participants. This study was registered at the University Hospital Medical Network as #000037771. Non-pregnant or pregnant women with RPL and pregnant women with FGR, HDP, or PD ≤ 34 GWs during their previous and/or current pregnancies who visited one of five centers between August 2019 and December 2021 underwent serum anti-β2GPI/HLA-DR antibody level measurements.

In this study, RPL was defined as the occurrence of two or more consecutive pregnancy losses according to the definition of the American Society for Reproductive Medicine [25]. FGR was defined as an estimated fetal body weight ≤ the mean −1.5 standard deviation for GW. HDP was defined as hypertension during pregnancy with or without proteinuria, organ damage, or uteroplacental dysfunction. Hypertension was defined as a maternal systolic blood pressure (BP) of ≥140 mmHg and/or a diastolic BP of ≥90 mmHg on two or more occasions at least 4 h apart.

Conversely, pregnant women who delivered full-term normal-birth-weight newborns but who had neither autoimmune diseases nor adverse obstetric outcomes, including RPL, FGR, HDP, or PD, in their previous and current pregnancies also underwent serum anti-β2GPI/HLA-DR antibody measurements after birth, and they served as controls.

The clinical data of all participants, including maternal age; gravidity and parity; BMI before pregnancy; prior and current maternal complications, including thrombosis, autoimmune diseases, and chronic hypertension; use of immunosuppressive drugs (i.e., PSL, azathioprine, tacrolimus, etc.); smoking habit; a history of ART therapy; prior or current obstetric adverse outcomes; gestational age at delivery; birth weight; and sex of newborns, were collected.

Furthermore, to identify causal and risk factors for RPL, women with RPL received checkups, including ultrasonography, to detect uterine malformations; serum tests of thyroid function (e.g., thyroid-stimulating hormone and free thyroxine were measured using an Architect i2000SR Analyzer [Abbott, Tokyo, Japan]); chromosomal karyotyping (Special References Laboratories [SRL], Tokyo, Japan) of peripheral blood from women and their spouses; measurements of serum aPL levels (e.g., lupus anticoagulant [SRL, Tokyo, Japan], Ig G, and M aCL, IgG/M aβ2GPI, and β2GPI-dependent aCL antibody [a Quanta Flash Antiphospholipid Assay Panel {Inova Diagnostics, San Diego, CA, USA}]), and hemostatic molecular markers (e.g., protein S, protein C, and coagulation factor XII activity levels [SRL, Tokyo, Japan]). Data obtained from these checkups were also collected.

Anti-β2GPI/HLA-DR antibody serum levels were measured at HuLA Immune Inc. (Tokyo, Japan), currently Revorf Co., Ltd. (Tokyo, Japan), using previously described methods [12]. cDNA was prepared from pooled human peripheral blood mononuclear cells (3H Biomedical, Uppsala, Sweden) and cloned into pME18S or pCAGGS expression vectors. Polyethylenimine Max reagent (PolyScience, Valley Road, Warrington, PA, USA) was used for transient transfection. To generate GFP-labeled β2GPI/HLA-DR-expressing cells or DsRed-labeled HLA-DR-expressing cells, 293T cells were co-transfected with GFP, β2GPI, and HLA–DRA*01:01 and DRB1*07:01 or with DsRed and HLA–DRA*01:01 and DRB1*07:01. Each transfectant was aliquoted (3 × 10^6^ cells/tube) with the cryoprotectant medium (Cell Banker 1 plus, Takara, Kusatsu, Japan) and stored at −80 °C until used. The mean fluorescence intensity (MFI) of IgG binding to these transfected cells in the sample sera was analyzed using flow cytometry (FACSLyric, Becton Dickinson, Franklin Lakes, NJ, USA). Both transfected cells were simultaneously incubated with each serum sample in 96-well plates (10^2^-fold dilution, 20 μL, for 30 min), followed by a 20-min incubation with APC-labeled anti-human IgG (Jackson ImmunoResearch, West Grove, PA, USA). Specific IgG binding to the β2GPI/HLA-DR complex was calculated by subtracting the MFI of IgG binding to cells transfected with HLA-DR alone from cells transfected with both β2GPI and HLA-DR (Figure 2A–C). A serum sample from a woman with RPL in whom anti-β2GPI/HLA-DR antibodies are detectable after a 10^6^-fold dilution was used as a standard, defined as 1000 units (U) [12]. A serum from a woman with APS [11] was used for unit calibration. Anti-β2GPI/HLA-DR antibody levels in each serum sample were calculated from the standard curve, which was generated by measuring specific IgG binding to the β2GPI/HLA-DR complex in serially diluted standard serum (10^2^- to 10^6^-fold dilution) (Figure 2D). All measurements were performed in duplicate, and mean values were defined as the anti-β2GPI/HLA-DR antibody level of the sample. The normal range of the anti-β2GPI/HLA-DR antibody level (<73.3 U) was determined using sera from 374 healthy controls, including 300 women and 74 men, with a 99th percentile value after outlier removal.

Clinical characteristics were compared between each group of women who had a specific adverse obstetric outcome and the control women. Differences between the two groups were analyzed using the Student *t*-test, Fisher exact test, and the χ^2^ test. The crude odds ratios (cORs) and adjusted ORs (aORs), which were adjusted for maternal age, BMI (kg/m^2^), and smoking habit, of each adverse obstetric outcome for anti-β2GPI/HLA-DR antibody positivity (≥73.3 U) were determined using univariate and multivariable logistic analyses, respectively. Statistical significance was set at a *p*-value of < 0.05. All statistical analyses were performed using EZR ver. 1.61 (Saitama Medical Center, Jichi Medical University, Saitama, Japan), a graphical user interface for R (The R Foundation for Statistical Computing, Vienna, Austria).

## 5. Patents

K.T. and H.A. hold a patent related to autoantibody detection (patent no. WO2015008498). No other disclosures relevant to this article were reported.

## Figures and Tables

**Figure 1 ijms-24-10958-f001:**
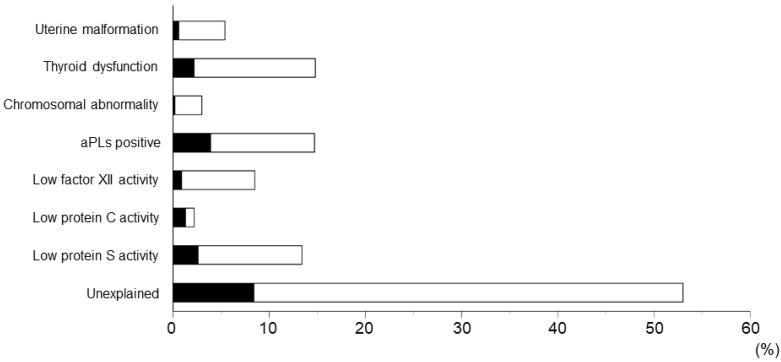
Risk factors for recurrent pregnancy loss (RPL) among 462 women with RPL. Black bars indicate the frequencies of women with RPL who tested positive for anti-β2GPI/HLA-DR antibody (*n* = 78). Abbreviations: aPLs, antiphospholipid antibodies; β2GPI, β2-glycoprotein I; HLA, human leukocyte antigen.

**Figure 2 ijms-24-10958-f002:**
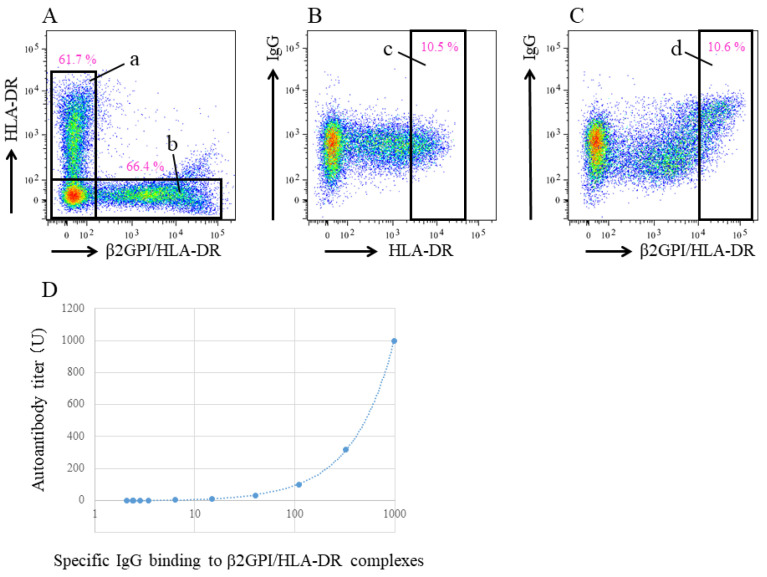
Quantification method for the autoantibody against β2GPI/HLA-DR complex. IgG binding to both β2GPI/HLA-DR complex-expressing cells (GFP-labeled) and HLA-DR-expressing cells (Ds-Red labeled) were analyzed using flow cytometry. Cells are gated on Ds-Red positive cells (**A**, a) or GFP positive cells (**A**, b). Specific IgG binding to the β2GPI/HLA-DR complex is calculated by subtracting the MFI of IgG binding to Ds-Red bright cells (**B**, c) from the MFI of IgG binding to GFP bright cells (**C**, d). A standard curve is generated from the specific IgG binding to β2GPI/HLA-DR complexes in sequentially diluted standard sera (**D**).

**Table 1 ijms-24-10958-t001:** Clinical characteristics and anti-β2GPI/HLA-DR antibody prevalence in women with RPL and women in the control group.

	RPL *n* = 462	Control *n* = 488	*p*-Value
Age (years)	35.5 ± 4.7	33.6 ± 5.3	<0.001
BMI (kg/m^2^)	21.5 ± 3.5	22.5 ± 4.4	<0.001
Gravidity (no)	3.7 ± 1.7	1.9 ± 1.0	<0.001
Parity (no)	0.5 ± 0.7	0.8 ± 0.8	<0.001
History of spontaneous miscarriages (no)	2.8 ± 1.4	0.1 ± 0.3	<0.001
History of stillbirths at 22 GWs or later (no)	0.05 ± 0.3	0.01 ± 0.09	<0.001
History of thrombosis	4 (0.9%)	0 (0%)	0.06
Autoimmune diseases	34 (7.4%)	0 (0%)	<0.001
Use of immunosuppressive drugs	9 (1.9%)	0 (0%)	<0.01
Chronic hypertension	2 (0.4%)	0 (0%)	0.2
Smoking habit	41 (8.9%)	40 (8.2%)	0.8
Anti-β2GPI/HLA-DR antibody positive	78 (16.9%)	27 (5.5%)	<0.001

Data are expressed as means ± standard deviations or numbers (percentages). Abbreviations: β2GPI, β2-glycoprotein I; HLA, human leukocyte antigen; RPL, recurrent pregnancy loss; BMI, body mass index; no, number; GWs, gestational weeks.

**Table 2 ijms-24-10958-t002:** Comparison of clinical characteristics between anti-β2GPI/HLADR antibody-positive and anti-β2GPI/HLADR antibody-negative women with RPL.

	Total *n* = 462	Anti-β2GPI/HLA-DR–Positive Women with RPL *n* = 78	Anti-β2GPI/HLA-DR–Negative Women with RPL *n* = 384	*p*-Value
Age (years)	35.5 ± 4.7	34.9 ± 4.9	35.7 ± 4.6	0.2
BMI (kg/m^2^)	21.5 ± 3.5	22.2 ± 3.9	21.4 ± 3.4	0.08
Gravidity (no)	3.7 ± 1.7	3.4 ± 1.4	3.7 ± 1.7	0.1
Parity (no)	0.5 ± 0.7	0.4 ± 0.6	0.5 ± 0.7	0.3
History of spontaneous miscarriages (no)	2.8 ± 1.4	2.5 ± 1.3	2.8 ± 1.4	0.06
History of stillbirths at 22 GWs or later (no)	0.1 ± 0.3	0.1 ± 0.4	0.0 ± 0.2	0.2
History of thrombosis	4 (0.9%)	2 (2.6%)	2 (0.5%)	0.1
Autoimmune diseases	34 (7.4%)	15 (19.2%)	19 (4.9%)	<0.001
Use of immunosuppressive drugs	9 (1.9%)	4 (5.1%)	5 (1.3%)	<0.05
Chronic hypertension	2 (0.4%)	0 (0%)	2 (0.5%)	1.0
Smoking habit	41 (8.9%)	8 (10.3%)	33 (8.6%)	0.6
Uterine malformation	25 (5.4%)	3 (3.8%)	22 (5.7%)	0.8
Thyroid dysfunction	68 (14.7%)	10 (12.8%)	58 (15.1%)	0.7
Chromosomal abnormality	14 (3.0%)	1 (1.3%)	13 (3.4%)	0.5
aPL positive	68 (14.7%)	18 (23.1%)	50 (13.0%)	<0.05
Low factor XII activity	39 (8.4%)	4 (5.1%)	35 (9.1%)	0.4
Low protein C activity	10 (2.2%)	6 (7.7%)	4 (1.0%)	0.4
Low protein S activity	62 (13.4%)	12 (15.4%)	50 (13.0%)	0.7
Unexplained	245 (53.0%)	39 (50.0%)	206 (53.6%)	0.6

Data are expressed as means ± standard deviations or numbers (percentages). Abbreviations: β2GPI, β2-glycoprotein I; HLA, human leukocyte antigen; RPL, recurrent pregnancy loss; BMI, body mass index; no, number; GWs, gestational weeks; aPL, antiphospholipid antibody.

**Table 3 ijms-24-10958-t003:** Clinical characteristics and anti-β2GPI/HLA-DR antibody prevalence in women with FGR and women in the control group.

	FGR			Control *n* = 488
	Previous *n* = 49	Current *n* = 83	All *n* = 124	
Age (years)	33.6 ± 4.1	31.6 ± 5.2 **	32.4 ± 5.0 *	33.6 ± 5.3
BMI (kg/m^2^)	22.3 ± 4.4	23.3 ± 5.0	22.8 ± 4.6	22.5 ± 4.4
Gravidity (no)	2.7 ± 1.2 **	2.1 ± 1.2	2.3 ± 1.3 **	1.9 ± 1.0
Parity (no)	1.2 ± 0.8 **	0.7 ± 0.9	0.9 ± 0.9	0.8 ± 0.8
History of spontaneous miscarriages (no)	0.4 ± 0.8 **	0.3 ± 0.7 **	0.3 ± 0.7 **	0.1 ± 0.3
History of stillbirths at 22 GWs or later (no)	0.1 ± 0.3 *	0	0.05 ± 0.2	0.01 ± 0.09
History of thrombosis	0 (0%)	0 (0%)	0 (0%)	0 (0%)
Autoimmune diseases	5 (10.2%) **	4 (4.8%) **	7 (5.7%) **	0 (0%)
Use of immunosuppressive drugs	1 (2.0%)	2 (2.4%) *	2 (1.6%) *	0 (0%)
Chronic hypertension	2 (4.0%) **	2 (2.4%) *	3 (2.4%) **	0 (0%)
Smoking habit	1 (2.0%)	6 (7.2%)	7 (5.7%)	40 (8.2%)
Pregnancy following ART	7 (14.3%)	8 (9.6%)	13 (10.5%)	75 (15.3%)
Anti-β2GPI/HLA-DR antibody positive	9 (18.4%) **	12 (14.5%) **	19 (15.3%) **	27 (5.5%)

Data are expressed as means ± standard deviations or numbers (percentages). * and ** indicate that significant differences are noted between the FGR and control groups with *p*-values of <0.05 and <0.01, respectively. Abbreviations: β2GPI, β2-glycoprotein I; HLA, human leukocyte antigen; FGR, fetal growth restriction; BMI, body mass index; no, number; GWs, gestational weeks; ART, assisted reproductive technology.

**Table 4 ijms-24-10958-t004:** Clinical characteristics and anti-β2GPI/HLA-DR antibody prevalence in women with HDP and control women.

	HDP			Control *n* = 488
	Previous *n* = 64	Current *n* = 79	All *n* = 138	
Age (years)	35.4 ± 4.3 **	35.0 ± 5.6 *	35.1 ± 5.0 **	33.6 ± 5.3
BMI (kg/m^2^)	23.2 ± 4.7	24.8 ± 6.5 **	24.0 ± 5.9 *	22.5 ± 4.4
Gravidity (no)	2.3 ± 1.2 *	1.9 ± 1.0	2.0 ± 1.1	1.9 ± 1.0
Parity (no)	1.0 ± 0.8	0.6 ± 0.7	0.8 ± 0.8	0.8 ± 0.8
History of spontaneous miscarriages (no)	0.4 ± 0.8 **	0.3 ± 0.6 *	0.3 ± 0.7 **	0.1 ± 0.3
History of stillbirths at 22 GWs or later (no)	0.06 ± 0.2	0.02 ± 0.1	0.04 ± 0.2	0.01 ± 0.09
History of thrombosis	1 (1.6%)	0 (0%)	1 (0.7%)	0 (0%)
Autoimmune diseases	3 (4.7%) **	6 (7.6%) **	9 (6.5%) **	0 (0%)
Use of immunosuppressive drugs	1 (1.6%)	3 (3.8%) **	4 (2.9%) **	0 (0%)
Chronic hypertension	10 (15.6%) **	17 (21.5%) **	24 (17.4%) **	0 (0%)
Smoking habit	1 (1.6%)	13 (16.5%) *	14 (10.1%)	40 (8.2%)
Pregnancy following ART	16 (25.0%)	14 (17.7%)	28 (20.3%)	75 (15.3%)
Anti-β2GPI/HLA-DR antibody positive	12 (18.8%) **	13 (16.5%) **	24 (17.4%) **	27 (5.5%)

Data are expressed as means ± standard deviations or numbers (percentages). * and ** indicate that significant differences are noted between the HDP and control groups with *p*-values of <0.05 and <0.01, respectively. Abbreviations: β2GPI, β2-glycoprotein I; HLA, human leukocyte antigen; HDP, hypertensive disorders of pregnancy; BMI, body mass index; no, number; GWs, gestational weeks; ART, assisted reproductive technology.

**Table 5 ijms-24-10958-t005:** Clinical characteristics and anti-β2GPI/HLA-DR antibody prevalence in women with PD before 34 GWs and women in the control group.

	Preterm Delivery ≤ 34 GW			Control *n* = 488
	Prior *n* = 35	Current *n* = 39	All *n* = 71	
Age (years)	35.3 ± 4.3 *	34.4 ± 4.8	34.9 ± 4.6 *	33.6 ± 5.3
BMI (kg/m^2^)	21.7 ± 2.7	22.7 ± 2.9	22.2 ± 2.9	22.5 ± 4.4
Gravidity (no)	2.7 ± 1.3 **	2.3 ± 1.0 **	2.5 ± 1.2 **	1.9 ± 1.0
Parity (no)	1.3 ± 0.8 **	1.0 ± 0.8	1.1 ± 0.8 **	0.8 ± 0.8
History of spontaneous miscarriages (no)	0.5 ± 1.1	0.4 ± 0.8 *	0.5 ± 0.9 **	0.1 ± 0.3
History of stillbirths at 22 GWs or later (no)	0.1 ± 0.3	0.03 ± 0.2	0.07 ± 0.3	0.01 ± 0.09
History of thrombosis	0 (0%)	0 (0%)	0 (0%)	0 (0%)
Autoimmune diseases	3 (8.6%) **	2 (5.1%) **	5 (7.0%) **	0 (0%)
Use of immunosuppressive drugs	2 (5.7%) **	0 (0%)	2 (2.8%) *	0 (0%)
Chronic hypertension	3 (8.6%) **	1 (2.6%)	4 (5.6%) **	0 (0%)
Smoking habit	5 (14.3%)	5 (12.8%)	10 (14.1%)	40 (8.2%)
Pregnancy following ART	5 (14.3%)	5 (12.8%)	10 (14.1%)	75 (15.3%)
Anti-β2GPI/HLA-DR antibody positive	5 (14.3%)	3 (7.7%)	8 (11.3%)	27 (5.5%)

Data are expressed as means ± standard deviations or numbers (percentages). * and ** indicate that significant differences are noted between the PD ≤ 34 GWs and control groups with *p*-values of <0.05 and <0.01, respectively. Abbreviations: β2GPI, β2-glycoprotein I; HLA, human leukocyte antigen; GWs, gestational weeks; BMI, body mass index; no, number; ART, assisted reproductive technology.

**Table 6 ijms-24-10958-t006:** Crude and adjusted odds-ratios of anti-β2GPI/HLA-DR positivity for each adverse obstetric outcome.

Adverse Obstetric Outcome		Crude OR	95% CI	*p*-Value	Adjusted OR *	95% CI	*p*-Value
RPL	*n* = 462	3.5	2.2–5.5	<0.001	3.3	1.9–5.6	<0.001
FGR							
Previous	*n* = 49	3.8	1.7–8.7	<0.005	3.1	1.2–8.0	<0.05
Current	*n* = 83	2.9	1.4–6.0	<0.005	2.6	1.2–5.6	<0.05
All	*n* = 124	3.1	1.7–5.8	<0.001	2.7	1.3–5.3	<0.01
HDP							
Previous	*n* = 64	3.9	1.9–8.2	<0.001	2.5	1.01–6.2	<0.05
Current	*n* = 79	3.4	1.7–6.8	<0.001	2.7	1.2–6.0	<0.05
All	*n* = 138	3.6	2.0–6.5	<0.001	2.7	1.4–5.3	<0.01
PD ≤ 34 GWs							
Previous	*n* = 35	2.8	1.02–7.9	<0.05	2.9	0.9–9.2	0.08
Current	*n* = 39	1.4	0.4–5.0	0.6	1.7	0.5–6.0	0.4
All	*n* = 71	2.2	0.9–5.0	0.07	2.3	0.9–5.7	0.08

ORs were obtained using univariate or multivariable logistic regression analyses. * Adjusted for maternal age, body mass index (kg/m^2^), and smoking habit. Abbreviations: β2GPI, β2-glycoprotein I; HLA, human leukocyte antigen; OR, odds ratio; CI, confidence interval; RPL, recurrent pregnancy loss; FGR, fetal growth restriction; HDP, hypertensive disorders of pregnancy; PD, preterm delivery; GWs, gestational weeks.

## Data Availability

Not applicable.

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
