# Peer review of "Anti-β2-glycoprotein I/HLA-DR Antibody and Adverse Obstetric Outcomes"

_ijms, 2023, doi:10.3390/ijms241310958_

Round 1
Reviewer 1 Report
Thank you for your submission of the manuscript “Anti-beta2-glycoprotein I/HLA-DR antibody and adverse obstetric outcomes” by Kenji Tanimura et al. evaluated the presence of Anti-beta2-glycoprotein I/HLA-DR antibody in relation to the risk factors for and adverse obstetric outcomes in a cohort of Japan APS patients.
The study deals with a topic that is important both for clinical practice and has a scientific contribution. The large sample size is impressive, allowing comparisons to be made with a control group.
A suggestion, if feasible, to add some clinical information. For example, did the patients only have obstetric APS, or did they also have a history of thrombosis, for example?
What exactly are the "other autoimmune diseases" in the “34 with autoimmune disorders” mentioned patients? What is their therapeutic treatment?
If, for example, it is about SLE, what kind of specific therapeutic treatment was applied? Maybe the therapy has an influence on the result (obstetric outcomes)?
The methodology is quality.
Author Response
Point-by-point response to reviewers’ comments
Reviewer 1
Overall comment: Thank you for your submission of the manuscript “Anti-beta2-glycoprotein I/HLA-DR antibody and adverse obstetric outcomes” by Kenji Tanimura et al. evaluated the presence of Anti-beta2-glycoprotein I/HLA-DR antibody in relation to the risk factors for and adverse obstetric outcomes in a cohort of Japan APS patients. The study deals with a topic that is important both for clinical practice and has a scientific contribution. The large sample size is impressive, allowing comparisons to be made with a control group.
Response: We thank the reviewer for quite positively evaluating our findings and providing valuable comments. However, we want the reviewer to understand that our present study included not only APS patients but also non-APS patients.
Comment 1: A suggestion, if feasible, to add some clinical information. For example, did the patients only have obstetric APS, or did they also have a history of thrombosis, for example?
Response: We thank the reviewer for providing a valuable comment. According to the reviewer’s suggestion, we added the number and proportion of women who had a history of thrombosis in Table 1, 2, 3, 4, and 5.
Along with this revision,
The sentence in Lines 99–101 was changed to “The proportions of women who had a history of thrombosis, chronic hypertension, and a smoking habit were not different between the two groups.”,
and
the sentence in Lines 156–159 was changed to “Maternal BMI and the proportion of women who had a history of thrombosis, a smoking habit and pregnancies following assisted reproductive….”,
and
the sentence in Lines 199–203 was changed to “Parity, the number of prior stillbirths at ≥ 22 GWs, and the proportion of women who had a history of thrombosis and pregnancies following ART were….”,
and in Lines 245–249, the sentence was changed to “Maternal BMI, number of prior stillbirths at ≥ 22 GWs, and the proportion of women who had a history of thrombosis, a smoking habit, and pregnancies….”.
Comment 2: What exactly are the "other autoimmune diseases" in the “34 with autoimmune disorders” mentioned patients? What is their therapeutic treatment?
If, for example, it is about SLE, what kind of specific therapeutic treatment was applied? Maybe the therapy has an influence on the result (obstetric outcomes)?
Response: We thank the reviewer for providing important suggestions. As I mentioned before, our present study included not only APS patients but also non-APS patients.
According to the reviewer’s suggestion, in Lines 92–94, “Autoimmune diseases of the 34 women in the RPL group were as follows: APS, n=13; Graves’ disease, n=7; systemic lupus erythematosus (SLE), n=5; Sjögren syndrome (SJS), n=4; RA, n=3; dermatomyositis, n=1; and autoimmune hepatitis, n=1.” was added.
In addition, we added the number and proportion of women who used immunosuppressive drugs in Table 1, 2, 3, 4, and 5.
As the reviewer pointed out, all SLE patients were treated with oral prednisolone (PSL) with or without another immunosuppressant (i.e., azathioprine, tacrolimus, etc.). But, unfortunately, it is unclear whether these therapies influenced on the prevalence of adverse obstetric outcomes and positive rate for anti-β2GPI/HLA-DR antibody, because the number of patients who used immunosuppressive drugs in this study was small.
Along with this revision,
the sentence in Lines 442–447 in the Materials and Methods section was changed to “The clinical data of all participants, including maternal age; gravidity and parity; BMI before pregnancy; prior and current maternal complications, including thrombosis, autoimmune diseases and chronic hypertension; use of immunosuppressive drugs (i.e., PSL, azathioprine, tacrolimus, etc.); smoking habit;….”,
and
the sentence in Lines 94–97 in the Results section was changed to “The RPL group had significantly higher maternal age, gravidity, number of prior spontaneous miscarriages and prior stillbirths at ≥ 22 GWs, and proportion of women with autoimmune diseases and women who used immunosuppressive drugs than the control group.”,
and
the sentence in Lines 126–129 in the Results section was changed to “The proportions of patients who had autoimmune diseases, who used immunosuppressive drugs, and who were positive for aPLs…”,
and
the sentence in Lines 148–150 in the Results section was changed to “Maternal age was significantly lower and the proportion of women who used immunosuppressive drugs was significantly higher in women who had FGR during their current pregnancies than that in the control group.”
and
the sentence in Lines 197–199 in the Results section was changed to “The group of women who had HDP during their current pregnancies had significantly higher maternal BMI and proportion of patients who used immunosuppressive drugs and who had a smoking habit than the control group.”
and
the sentence in Lines 241–243 in the Results section was changed to “Women who had PD ≤ 34 GWs during their previous pregnancies had significantly higher maternal age, parity, and proportion of women who used immunosuppressive drugs than those in the control group.”
and
in Lines 340–347 in the Discussion section, “On the other hand, some patients in the groups of RPL, FGR, HDP, and PD ≤ 34 GWs used immunosuppressive drugs for treatments of underlying autoimmune diseases. All of them were treated with oral prednisolone (PSL), and a few of them were treated with oral PSL combined with another immunosuppressant (i.e., azathioprine, tacrolimus, etc.). However, it is unclear that these immunosuppressive agents influence the production of anti-β2GPI/HLA-DR antibodies. The number of patients who used immunosuppressive drugs in this study was small; therefore, we believe that the results of this study were not influenced by the use of these drugs.” was added.
Comment 3: The methodology is quality.
Response: We thank the reviewer for giving a positive evaluation.

Reviewer 2 Report
This manuscript describes a prospective, multicenter, cross-sectional study and investigating the association between anti-beta-2-GPI/HLA-DR antibodies and adverse obstetric outcomes as well as recurrent pregnancy loss (RPL). Study included 462 women with RPL and 488 controls. The authors found that nearly 17% of the RPL group were positive for anti-beta-2-GPI/HLA-DR antibodies, compared to only 5.5% of the controls. Clinical details of the patients are well described. This work is important because of the newly identified autoantibody. The essence fo the findings are that anti-beta-2-GPI/HLA-DR antibodies are found in a statistically significant proportion of women with adverse pregnancy outcomes. I have the following comments/criticisms:
1. The authors state that the study is prospective in nature. Do they mean prospective with regard to a current pregnancy, or do they mean that they collected blood for this study as patients came in and provided a history? Did all patients in the study have a prospectively followed pregnancies after their initial autoantibody determinations? If so, how many prospectively followed pregnancies are included. The best study design would report and analyze outcomes in the first prospectively-followed pregnancy following the initial positive or negative autoantibody test result.
2. Were any of the patients being treated with him you know modulatory medications? I would guess that some were. These details are very important and must be accounted for.
3. Autoantibody test results confirmed by repeat testing? This should be made clear.
4. Any of the patients included in the study have APS. This should be made clear and accounted for in the analyses. It would be quite interesting to better understand outcomes related to anti-beta-2-GPI/HLA-DR antibodies in the absence of traditional antiphospholipid antibodies.
5. I am not certain that the authors have clearly shown that anti-beta-2-GPI/HLA-DR antibodies are associated with adverse pregnancy outcomes after adjustment for several variables, not made clear in the manuscript. Please include treatment during pregnancy, history of autoimmune disease, history of chronic, hypertension, and others
Author Response
Point-by-point response to reviewers’ comments
Reviewer 2
Overall comment: This manuscript describes a prospective, multicenter, cross-sectional study and investigating the association between anti-beta2-GPI/HLA-DR antibodies and adverse obstetric outcomes as well as recurrent pregnancy loss (RPL). Study included 462 women with RPL and 488 controls. The authors found that nearly 17% of the RPL group were positive for anti-beta2-GPI/HLA-DR antibodies, compared to only 5.5% of the controls. Clinical details of the patients are well described. This work is important because of the newly identified autoantibody. The essence of the findings are that anti-beta2-GPI/HLA-DR antibodies are found in a statistically significant proportion of women with adverse pregnancy outcomes. I have the following comments/criticisms.
Response: We thank the reviewer for quite positively evaluating our findings and providing valuable comments.
Comment 1: The authors state that the study is prospective in nature. Do they mean prospective with regard to a current pregnancy, or do they mean that they collected blood for this study as patients came in and provided a history? Did all patients in the study have a prospectively followed pregnancies after their initial autoantibody determinations? If so, how many prospectively followed pregnancies are included. The best study design would report and analyze outcomes in the first prospectively-followed pregnancy following the initial positive or negative autoantibody test result.
Response: We thank the reviewer for providing a critical suggestion. This study is not a prospective cohort study but a prospective cross-sectional study where patients who visited our centers and had adverse obstetric outcomes during prior and/or current pregnancies underwent blood sampling. But we have already undertook another prospective cohort study to prove the accurate pathogenicity of anti-β2GPI/HLA-DR antibody.
In Lines 392–397 in the Discussion section, “The present study had some limitations. First, this study is not a prospective cohort study but a prospective cross-sectional study; therefore, we could not prove the accurate pathogenicity of anti-β2GPI/HLA-DR antibody. Presently, a cohort study, where normal pregnant women undergo serum anti-β2GPI/HLA-DR antibody level measurements in the first trimester and they are followed up for the occurrence of adverse obstetric outcomes, is already being undertaken.” was added.
Comment 2: Were any of the patients being treated with him you know modulatory medications? I would guess that some were. These details are very important and must be accounted for.
Response: We thank the reviewer for providing a critical suggestion. Another reviewer asked us a similar question.
According to the reviewers’ suggestion,
the sentence in Lines 442–447 in the Materials and Methods section was changed to “The clinical data of all participants, including maternal age; gravidity and parity; BMI before pregnancy; prior and current maternal complications, including thrombosis, autoimmune diseases, and chronic hypertension; use of immunosuppressive drugs (i.e., PSL, azathioprine, tacrolimus, etc.); smoking habit;….”,
and
we added the number and proportion of women who used immunosuppressive drugs in Table 1, 2, 3, 4, and 5.
and
in Lines 340–347 in the Discussion section, “On the other hand, some patients in the groups of RPL, FGR, HDP, and PD ≤ 34 GWs used immunosuppressive drugs for treatments of underlying autoimmune diseases. All of them were treated with oral prednisolone (PSL), and a few of them were treated with oral PSL combined with another immunosuppressant (i.e., azathioprine, tacrolimus, etc.).” was added.
Comment 3: Autoantibody test results confirmed by repeat testing? This should be made clear.
Response: We thank the reviewer for providing a critical suggestion.
As the reviewer pointed out, in the diagnostic criteria for APS, positive results from aPLs testing on 2 tests 12 months or more apart are required. Unfortunately, positive results for anti-β2GPI/HLA-DR antibody were not confirmed by repeat testing. Further study is needed to investigate whether anti-β2GPI/HLA-DR antibody positivity can persist.
In Lines 408–411 in the Discussion section about the limitation of our study, “Third, in the diagnostic criteria for APS, positive results from aPLs testing on 2 tests 12 months or more apart are required. However, repeat testing to confirm persistent positive results for anti-β2GPI/HLA-DR antibody was not performed in this study.” was added.
Comment 4: Any of the patients included in the study have APS. This should be made clear and accounted for in the analyses. It would be quite interesting to better understand outcomes related to anti-beta-2-GPI/HLA-DR antibodies in the absence of traditional antiphospholipid antibodies.
Response: We thank the reviewer for providing important suggestions.
According to the reviewers’ suggestion,
in Lines 379–382 in the Discussion section, “Indeed, 71 (91.0%) of 78 anti-β2GPI/HLA-DR antibody-positive women with PRL, 17 (89.5%) of 19 the antibody-positive women with FGR, 22 (91.7%) of 24 the antibody-positive women with HDP were not diagnosed with APS.” was added.
Comment 5: I am not certain that the authors have clearly shown that anti-beta-2-GPI/HLA-DR antibodies are associated with adverse pregnancy outcomes after adjustment for several variables, not made clear in the manuscript. Please include treatment during pregnancy, history of autoimmune disease, history of chronic hypertension, and others.
Response: We thank the reviewer for providing a critical suggestion.
We added the number and proportion of women who used chronic hypertension in Table 1, 2, 3, 4, and 5.
According to the reviewers’ suggestion,
the sentence in Lines 442–447 in the Materials and Methods section was changed to “prior and current maternal complications, including thrombosis, autoimmune diseases, and chronic hypertension; use of immunosuppressive drugs (i.e., PSL, azathioprine, tacrolimus, etc.); smoking habit;….”.
In our study, adjusted ORs were determined by multivariable logistic analyses where all variables were compared between the groups of patients with adverse obstetric outcomes and the control group. Because no women in the control group had autoimmune diseases, chronic hypertension, and used immunosuppressive drugs, we could not calculate the ORs adjusted for the presence of autoimmune diseases, chronic hypertension, and use of immunosuppressive drugs.

Round 2
Reviewer 2 Report
None